# Premature Ovarian Insufficiency: Procreative Management and Preventive Strategies

**DOI:** 10.3390/biomedicines7010002

**Published:** 2018-12-28

**Authors:** Jennifer J. Chae-Kim, Larisa Gavrilova-Jordan

**Affiliations:** 1Department of Obstetrics and Gynecology, East Carolina University, Greenville, NC 27834, USA; chaej18@ecu.edu; 2Department of Obstetrics and Gynecology, Augusta University, Augusta, GA 30912, USA

**Keywords:** premature ovarian insufficiency, in vitro fertilization, donor oocyte, fertility preservation

## Abstract

Premature ovarian insufficiency (POI) is the loss of normal hormonal and reproductive function of ovaries in women before age 40 as the result of premature depletion of oocytes. The incidence of POI increases with age in reproductive-aged women, and it is highest in women by the age of 40 years. Reproductive function and the ability to have children is a defining factor in quality of life for many women. There are several methods of fertility preservation available to women with POI. Procreative management and preventive strategies for women with or at risk for POI are reviewed.

## 1. Introduction

Premature ovarian insufficiency (POI) is the loss of normal hormonal and reproductive function of ovaries in women before age 40 as the result of premature depletion of oocytes. POI is characterized by elevated gonadotrophin levels, hypoestrogenism, and amenorrhea, occurring years before the average age of menopause. Previously referred to as ovarian failure or early menopause, POI is now understood to be a condition that encompasses a range of impaired ovarian function, with clinical implications overlapping but not synonymous to that of physiologic menopause. Sequelae of POI include infertility, subfertility with intermittent ovarian function, devastating impact on emotional well-being, sexual dysfunction, vasomotor symptoms, bone loss, or increased risk of cardiovascular disease.

The prevalence of POI ranges between 0.3 and 1% of reproductive age women. The incidence of POI increases with age in reproductive-aged women: It occurs in approximately 1 in 1000 women under age 20, and by the age of 35 years the incidence increases to 1 in 250 women, and by the age of 40 years it is 1 in 100 women [1]. POI affects approximately 10 to 25% of women presenting with amenorrhea [2]. For those presenting with infertility, POI along with other types of ovulatory dysfunction are found to be the cause in nearly 20% of patients.

### Etiology and Clinical Presentation

The etiology of POI is diverse, including genetic, acquired, or iatrogenic causes. The most common genetic cause of POI is Turner syndrome, a sex chromosome disorder caused by a missing or partially missing X chromosome. Premutations of Fragile X syndrome are also highly associated with POI. Premutation carriers of Fragile X syndrome have CGG trinucleotide expansions of about 50 to 200 repeats. Various defects in transcription factors, steroidogenic enzymes, or gonadotrophic receptors may also play a role. Acquired causes of POI may have an autoimmune origin, resulting from autoimmune polyglandular syndromes (APS) type 1 or 2, or are associated with other autoimmune disorders. About 20% of POI women have been found to have concomitant autoimmune disorders, particularly of the thyroid, pancreas, and adrenals. Anti-oocyte autoantibodies may also be present. Lymphocytic autoimmune oophoritis results in lymphocytic infiltration and destruction of theca cells of secondary and antral follicles, but not of primordial follicles. Environmental exposure to toxins, tuberculosis or viral agents such as mumps or cytomegalovirus, are also thought to be possible causes of acquired POI. POI may be iatrogenic, in the case of cancer therapy involving gonadotoxic radiation, or pelvic surgery. The vast majority of POI, however, remains idiopathic.

The clinical presentation of POI reflects a spectrum of impaired ovarian function, and also depends on the etiology. Adolescent patients with Turner syndrome or permutation carriers of Fragile X syndrome may present with primary amenorrhea. The typical features of Turner syndrome include short stature, “shield” chest, webbed neck, or cubitus valgus. Premutation carriers of Fragile X syndrome tend not to manifest the phenotypical features of Fragile X syndrome seen in females, such as cognitive impairment, mood disorders, and socially avoidant behavior. Older patients may present with secondary amenorrhea or with vasomotor symptoms due to estradiol deficiency. The physical examination of women with acquired POI may be unremarkable or reveal symptoms of concomitant autoimmune disorders. Enlarged cystic ovaries are often reported in autoimmune oophoritis on transvaginal ultrasonography. Some women present with infertility as the only manifestation of POI. Lastly, a subset of women who anticipate iatrogenic POI as a result of future intervention may present with sufficient ovarian reserve, and the clinician can counsel these patients regarding the predicted symptoms of POI.

## 2. Procreative Management

Ovarian reserve is a term that refers to the “population of primordial follicles,” measuring the procreative capacity of the ovary [3]. The potential number of oocytes peaks at 6 to 7 million around 20 weeks gestational age, and falls precipitously until birth when there are 300,000 to 400,000 oocytes remaining [4]. The average reproductive lifespan is about 450 monthly ovulatory cycles, with approximately 1000 follicles remaining by menopause [3]. Anti-Mullerian hormone, or AMH, correlates with the number of primordial follicles and indirectly measures the ovarian reserve.

The degree of residual ovarian function in women with POI is variable. Despite reduced ovarian reserve, approximately 5 to 10% of women with POI may experience spontaneous ovulation, resulting in conception and live birth [5]. These pregnancies are not associated with an increased risk for obstetrical or neonatal adverse outcomes [6]. Those who experience spontaneous ovulation, however, also tend to have infrequent ovulation secondary to POI, leading to a significant increase in time to conception when compared to women with normal ovarian reserve. Furthermore, the majority of women with POI are affected by infertility and need fertility treatment. Women at risk for POI due to genetic predisposition, medical conditions, and gonadotoxic treatment, benefit from fertility preservation. Methods of fertility preservation, as well as experimental methods currently under investigation, are presented here.

### 2.1. In Vitro Fertilization with Autologous Versus Donor Oocytes or Embryos

In Vitro Fertilization (IVF) with autologous oocytes is an effective treatment for women with POI when the residual ovarian reserve is sufficient for ovarian stimulation. Women with POI require significantly higher doses of exogenous gonadotropins to initiate folliculogenesis. They commonly have a poor response to stimulation with only four or fewer follicles available for oocyte retrieval. Limited oocyte recovery results in fewer or no embryos available for transfer or cryopreservation. Women with POI may require multiple consecutive IVF cycles to achieve conception. Factors contributing to IVF failure include reduced ovarian reserve, advanced maternal age, suboptimal endometrial environment, as well as procedural complications (ovarian stimulation, embryo transfer). Pregnancy and live birth rates largely depend on the female’s age [3].

Women with undetectable AMH, a critical marker of follicular depletion and ovarian reserve, as well as clinical hypoestrogenism, benefit from IVF with donor oocytes. For women with POI due to a genetic cause who wish to avoid genetic disease transmission, IVF with donor oocytes may be a particularly interesting and promising option of assisted reproductive technology (ART). Oocytes from genetically related or unrelated young reproductive age females can be utilized. In the last decade, an effective oocyte vitrification (very rapid freezing) technique has been developed for oocyte cryopreservation. Numerous studies have reported high success rates using oocyte donation; approximately 50 to 60% of attempts, per embryo transfer, lead to live pregnancy and birth. Thus, frozen donor oocytes are now commercially available via private oocyte banks. It is typical for a commercial oocyte bank to limit the age of the donors to 32–34 years, as well as screen donors for physical and mental health. Pregnancy rate and live birth rates depend on the age of the oocyte donor. For women with iatrogenic POI due to cancer therapy, pregnancy rates may also depend on whether the patient sustained endometrial injury secondary to pelvic radiation.

With an increasing inventory of cryopreserved embryos from infertile couples treated with IVF, there are excessive embryos that are donated either directly to women with POI or anonymously via embryo banks. Women with POI and without a male partner may benefit from donor embryos; other options available to these women including oocyte cryopreservation (egg banking) if there is sufficient ovarian reserve or IVF utilizing donor sperm.

### 2.2. Cryopreservation of Oocytes or Embryos

For reproductive age women who anticipate gonadotoxic chemotherapy or pelvic radiation that causes acute POI, oocyte and/or embryo cryopreservation are the best fertility preservation options [3]. Oocyte and embryo cryopreservation are no longer considered experimental, according to the American Society for Reproductive Medicine (ASRM). This option is particularly important to consider in patients whose chemotherapy regimens include highly gonadotoxic alkylating agents. In particular, cyclophosphamide is highly toxic to the ovary in a dose-dependent manner [3]. Women with a known genetic predisposition for POI, such as Fragile X syndrome permutation carriers, also benefit from prophylactic oocyte cryopreservation before ovarian function declines [7]. Live birth rates for embryo cryopreservation are approximately 30–40%; rates for oocyte cryopreservation are only slightly lower [8]. Cryopreservation via vitrification has been shown to reduce the risk of ice crystal formation during freezing, and improve oocyte survival after thawing, as well as increase live birth rates [9]. Evidence based on randomized control trials and meta-analyses suggests that pregnancy rates after IVF or intracytoplasmic sperm injection (ICSI) with vitrified oocytes are similar to that for fresh oocytes; and evidence suggests there is no increase in chromosomal abnormalities, congenital anomalies, or development delay in offspring from pregnancies resulting from cryopreserved oocytes versus the general population [8,9].

### 2.3. Gestational or Surrogate Carrier

Women with POI can benefit from IVF with an autologous or donor embryo transfer to a gestational carrier or surrogate carrier, respectively. For instance, women with POI related to Turner syndrome (45X0) or women with mosaic 46X0/46XX karyotype are not only infertile or sub-fertile but they also carry an increased risk of aortic dissection or rupture during pregnancy. Given the increased risk of maternal mortality, Turner syndrome is considered a relative contraindication for pregnancy, and an absolute contraindication in the presence of a cardiac anomaly secondary to Turner syndrome [10]. These women have an indication for a gestational carrier. The genetic and intended parents should undergo psychosocial education, genetic screening, a full physical evaluation including laboratory testing, any other state-specific requirements, and legal counseling regarding the gestational agreement [11].

### 2.4. In Vitro Maturation

In vitro maturation (IVM) is a beneficial method of fertility preservation for reproductive age women at high risk of POI as a result of gonadotoxic treatment, particularly when delaying such treatment carries an unacceptable medical risk to the patient. It may also be indicated for patients with malignancy who are contraindicated to ovarian hyperstimulation, as well as for patients at risk for ovarian hyperstimulation syndrome [12]. Immature oocytes, or those not yet exposed to luteinizing hormone or human chorionic gonadotropin, are collected from unstimulated ovaries, then matured in a medium in vitro [12]. They mature from prophase I stage to metaphase II stage. This is an experimental technique, and there are no established criteria regarding the ideal timing of oocyte retrieval, aspiration technique or selection of culture media. Given the in vitro environment, oocytes tend to be fertilized via ICSI. Live births have been reported, however, the implantation and pregnancy rates are lower than that of traditional IVF with mature oocytes [13].

### 2.5. Ovarian Tissue Cryopreservation and Orthotopic Versus Heterotopic Transplantation

For prepubertal females with cancer, cryopreservation of ovarian tissue prior to gonadotoxic therapy is an experimental though promising method of fertility preservation [14]. It is also an option for postpubertal patients who cannot delay cancer treatment for emergency IVF. Prior to cancer therapy, the cortex of the ovary that contains ovarian follicles is dissected off and cryopreserved via slow freezing or vitrification. After successful cancer treatment or during complete remission, the cortical ovarian tissue can be thawed and transplanted to the pelvis (orthotopic transplantation onto the ovarian medulla or nearby peritoneal window) or extrapelvic subcutaneous tissues such as the forearm or abdomen (heterotopic transplantation) [15]. Orthotopic transplantation has the unique advantage of the possibility of natural fertilization, given that the cortical tissue is grafted in close proximity to the fallopian tube [16]. Ovarian function usually lasts up to five years after transplantation [3]. Successful pregnancies and live births after orthotopic transplantation have been reported, at rates of approximately 30% and 23 to 25%, respectively [17,18,19,20]. Heterotopic transplantation requires IVF after oocyte retrieval from the extrapelvic site, and two successful live births of twins have been reported [21,22]. In one case, the heterotopic ovarian tissue was removed after childbearing due to the patient’s history of ovarian cancer, and was successfully refrozen and xenotransplanted, with histology-proven presence of pre-antral follicles in the tissue graft [22].

### 2.6. Methods to Minimize Reseeding Risk: Xenotransplantation, Artificial Ovary, In Vitro Follicle Growth

The main safety concern regarding ovarian tissue transplantation is the possibility of reseeding tumor cells; this is especially true in blood-borne cancers such as leukemia [16,23]. Soares et al. described an isolation technique yielding ovarian follicles, initially taken from leukemia patients, with no evidence of malignant cells after three washes [24]. The isolated follicles were then xenotransplanted onto the peritoneum of severe combined immunodeficient (SCID) mice, followed for 6 months, then assessed for the presence of leukemia [25]. Shapira et al. recently reported a case report of a leukemia patient who underwent ovarian tissue cryopreservation during remission and before bone marrow transplantation. When the patient desired pregnancy, ovarian tissue fragments were xenotransplanted onto SCID mice and followed for 6 months to confirm no evidence of leukemia cells. The ovarian tissue was transplanted back to the patient, who underwent IVF and a live birth [26].

Another method to minimize the risk of reseeding cancer cells involves the artificial ovary. In a recent study, preantral murine follicles and murine ovarian cells were isolated and placed in a novel artificial ovary, or a fibrin scaffold [27]. The artificial ovary aims to provide an environment that allows for follicle or cell growth outside the extracellular matrix or stromal environment; this is thought to reduce the risk of reintroducing cancer cells. The follicles and cells were then auto-transplanted onto the peritoneum of the donor mice and later found to be viable [27].

In vitro follicle growth (IVG) is an experimental and exciting new method of growing immature ovarian follicles in vitro, to eliminate the risk of reintroducing cancer cells. It may also be applicable to prepubertal patients for whom hormone stimulation is not an appropriate strategy to obtain viable follicles [28]. The immature (primordial) follicle is isolated and grown in a three-dimensional culture. A recent study found that by using a two-step culture strategy that mimics the follicle’s change in environment as they mature in vivo, preantral follicles were matured to the antral stage. Afterwards, follicles undergo IVM (as described in Section 2.4.), resulting in viable metaphase II stage oocytes [29]. There is still, however, the concern that follicles in vitro do not under genomic imprinting and thereby do not undergo normal development [30]. Some research has found that the morphology of follicles is altered after cryopreservation, and in vitro growth was significantly delayed in follicles that were cryopreserved via vitrification versus fresh follicles [31].

### 2.7. In Vitro Activation

Women with POI have diminished but variable ovarian function. In vitro activation is a method of “activating” or recruiting residual follicles into the pool of primordial follicles, which can develop into mature oocytes. Studies have reported live births after activating residual follicles in vitro with Akt stimulators [32,33], or PTEN inhibitor [34,35], with subsequent ovarian tissue auto-transplantation or egg retrieval for IVF. For IVA protocols involving cryopreservation of ovarian tissue, a likely mechanism involved in follicle recruitment is that the ovariectomy or ovarian fragmentation disrupts Hippo signaling, leading to follicle growth and mature oocytes [32]. A recent study showed that almost half of the subjects were found to have residual follicles, based on histological analysis; after ovarian fragmentation and Akt stimulation, 45% of those subjects demonstrated multiple antral follicles [36]. These results are promising for patients with progressive ovarian dysfunction, reduced follicular pool or amount of cryopreservated tissue available for fertility preservation [35].

### 2.8. Oogonial Stem Cells and Artificial Gametogenesis

Recent research on oogonial stem cells (OSC), or germline stem cells thought to contribute to post-natal oocyte production, has challenged the belief that the mammalian ovary contains a fixed and finite number of oocytes. Several studies have reported the isolation of OSCs from adult rodent, mice, and human ovaries [37]. White et al. reported successful isolation of OSCs from fresh or cryopreserved human ovarian tissue [38]. In a microenvironment where OSCs interact with somatic ovarian cells, OSCs have been shown to generate follicles capable of forming oocytes; in rodents, healthy offspring have resulted. Human OSCs have been shown to form oocyte-like structures in a xenotransplant murine model [37,38,39]. Hayashi et al. showed embryonic stem cells and pluripotent stem cells were induced into primordial germ cell-like cells (PGCLCs) in vitro; primordial germ cells also give rise to oocytes [40]. These findings have the potential to revolutionize the field of reproductive endocrinology, particularly for women with infertility secondary to POI. For instance, OSCs could be preserved in advance of anticipated POI, then replaced into the ovary when the patient desires pregnancy [41].

However, the role OSCs or PGCLCs play in the female ovarian has not been fully elucidated. In spite of the presence of OSCs, ovarian function eventually declines, and menopause is inevitable. This may be because of how an aging ovarian microenvironment affects the role of OSCs in follicular development [42]. The function of the granulosa cells or other signaling processes in the ovarian microenvironment may change with age. It has also been suggested that OSCs remain quiescent, and are never activated [41]. In sum, it still remains unclear how stem cells can be utilized to improve reproductive function. There currently is no therapy involving OSCs proven to contribute to the development of functional human gametes, and further research is required [43].

### 2.9. Fertility-Sparing Surgery

For early-stage ovarian tumors with low malignant potential in reproductive age females, fertility-sparing surgery such as cystectomy or unilateral oophorectomy is the standard of care [8]. For patients planning to receive pelvic irradiation, follicle loss or damage is a certain complication. Exposure to 5 to 10 Gy is highly toxic to the ovary, resulting in significant follicle loss [3]. Surgical transposition of the ovaries, or oophoropexy, away from the radiation field has been shown to preserve ovarian function in patients undergoing pelvic radiation [44]. The ovaries are surgically transposed laterally to the upper abdomen by securing them to the abdominal wall. A method of medial transposition behind the uterus has been reported [45]. The optimal timing of oophoropexy is just before radiation therapy [8]. Due to the risk of radiation scatter, biopsies of ovarian tissue may be cryopreserved for future use in case of ovarian failure [18]. A literature review found that laparoscopic ovarian transposition in women under age 40 was associated with an ovarian function preservation rate of 88.6% [44]. Spontaneous pregnancies and life births have been reported after ovarian transposition.

### 2.10. Fertoprotective Agents

New research has looked into adjuvant “ferto-protective” agents that prevent chemotherapy-induced ovarian damage and follicle depletion [8,18,46]. Various chemotherapeutic agents such as alkylating agents (cyclophosphamide) or platinum-based agents (cisplatin) cause loss of primordial follicle reserve, leading to reduced ovarian function or iatrogenic POI [46]. The Alkylating Agent Dose (AAD) and Cyclophosphamide Equivalent Dose (CED) are metrics clinicians can use to estimate the risk of long-term adverse outcomes of these therapies [47]. Ferto-protective agents are meant to be adjuvant therapies that shield the follicle reserve from the gonadotoxic effects of chemotherapy. An important concern is whether these agents affect the efficacy of chemotherapy treatment [46].

Several agents have been studied for their potentially ferto-protective effect. Sphingosine-1-phosphate, or SIP, inhibits the sphingomyelin apoptotic pathway which plays a role in the apoptosis of ovarian follicles. Pretreatment with SIP has been shown to reduce gonadotoxic effects in primates and xenografted human ovarian tissue [48]. Given its mechanism of action, however, SIP may interfere with the apoptotic action of chemotherapeutic agents. Granulocyte Colony-Stimulating Factor, or G-CSF, has been shown to reduce follicle loss in a murine model after cisplatin therapy [49]. The mechanism of G-CSF is unclear, but it is thought to promote neovascularization that counteracts the ischemic effects of chemotherapy [46], as well as anti-apoptotic actions on the ovary [8]. AS101 is a tellurium-based compound with anti-inflammatory and anti-apoptotic effects, thought to modulate the phosphatidylinositol 3-kinase (PI3K)/PTEN/Akt pathway, which is thought to regulate oocyte and granulosa cell development [8]. Kalich-Philosoph et al. found that cyclophosphamide reduces ovarian reserve by activating primordial follicles in cyclophosphamide-treated mice [50]. This “burnout” effect of follicle activation, however, was decreased by co-administration with AS101; further, AS101 was found to have a synergistic interaction with cyclophosphamide [50]. Significant research has been conducted on the co-administration of GnRH agonists during cancer therapy, and results have been conflicting. Current evidence suggests it has limited ovarian gonadoprotective effect during chemotherapy [51].

New research has also looked into the method of chemotherapy delivery. Ahn et al. reported that nano-encapsulation of Arsenic trioxide, or As_2_O_3_, increases the efficacy of chemotherapy and decreases its gonadotoxic effects on the ovary [52]. Nanobin encapsulation of the arsenic, or NB (Ni, As), is thought to target As_2_O_3_ thereby reducing its field of impact. The study also reported a novel in vitro follicle toxicity assay that aims to predict the in vivo gonadotoxicity of chemotherapy agents. Per the toxicity assay, NB (Ni, As) was shown to have a lower effective dose and lower toxicity versus free arsenic. The toxicity assay may potentially be applied to other chemotherapies in order to guide cancer therapy to be more ferto-protective [52].

## 3. Conclusions

For those women with POI who suffer from infertility, several procreative managements, and preventive strategies to preserve fertility are available. Selection of the procreative management strategy is patient-specific, depending on the clinical context as well as the patient’s existing ovarian reserve. Despite insufficient or absent ovarian reserve, pregnancy and live births have been successfully demonstrated in even experimental methods of fertility preservation. The methods of fertility preservation, including investigational treatment options, discussed in this review are summarized in Figure 1.

IVF and embryo or oocyte cryopreservation are established methods with well-documented pregnancy and live birth outcomes. Ovarian tissue cryopreservation is also increasingly practiced. It should be noted that these established methods of fertility preservation are not readily available worldwide [53]. For instance, embryo cryopreservation is legally restricted in countries like Germany, Switzerland, and Austria; it was banned in Italy from 2004 to 2009 [54]. Embryo cryopreservation was restricted to married women until recently in the United Arab Emirates; it remains restricted to married couples in most provinces of China [55]. Many countries have a minimum or maximum age limit for cryopreservation, such as older than age 18 (Spain, France) or younger than 42 years (Malta, France) or 45 years (Slovenia), even for medically indicated cryopreservation [55]. In the United States (US), the regulations regarding cryopreservation are less strict; the data cited in Section 2.2 are from research studies or organizations based in the US. Restrictions aimed at “social egg freezing,” or cryopreservation for non-medical indications, may inadvertently affect women with POI who require the use of donor oocytes or embryos for ART.

Other treatment options are increasingly relevant as younger women and adolescents are being diagnosed with cancer, requiring therapy that may be highly gonadotoxic. Oophoropexy is a surgical intervention that has been shown to preserve ovarian function. “Ferto-protective” agents are adjuvant therapies being studied for their potential to protect against ovarian damage and follicle depletion during chemotherapy. Ovarian tissue cryopreservation is under investigation, as well as various methods designed to reduce the risk of reseeding cancer cells after thawing, such as xenotransplantation and in vitro follicle growth.

Significant research has gone into clarifying the role of oogonial stem cells in the post-natal ovary. Currently, there is no evidence of neo-oogenesis, and it remains unclear how oogonial stem cells contribute to the ovarian reserve. A breakthrough in this area would revolutionize the field of reproductive medicine. Other innovative treatment options being explored include in vitro activation, in which residual follicles are recruited into the available pool of ovarian follicles.

In summary, POI is a complex condition arising from numerous possible etiologies, with multiple sequelae, including infertility secondary to diminished ovarian reserve. Several established methods of fertility preservation are available to women with POI. Further research on experimental methods is underway, and the field of fertility preservation is rapidly expanding.

## Figures and Tables

**Figure 1 biomedicines-07-00002-f001:**
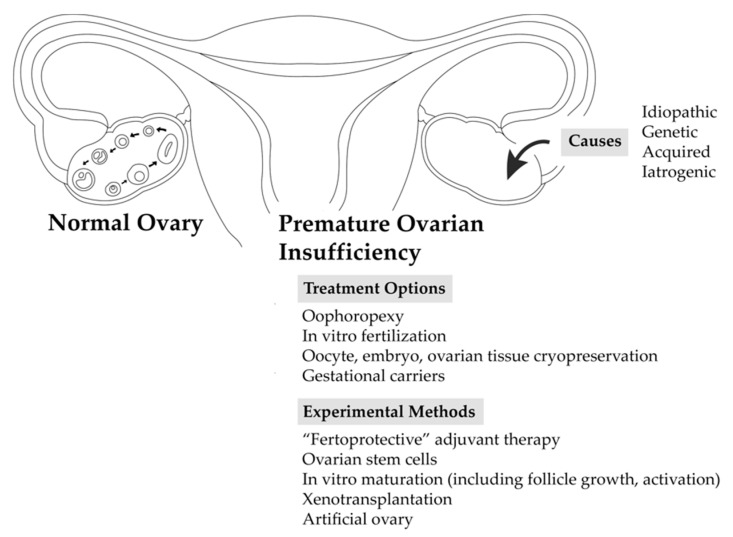
Fertility preservation options in premature ovarian insufficiency.

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
