# Peer review of "Premature Ovarian Insufficiency: Procreative Management and Preventive Strategies"

_biomedicines, 2018, doi:10.3390/biomedicines7010002_

Reviewer 1 Report

This review presents a sinottic view of the so far known causes of Premature Ovarian Insufficency (POI) as well as the clinical managment of POI patients. The subject is highly topical, the manuscript is well written and take into account of the several aspects belonging these pathologies. I have just two short comments:

1) in the "abstract" the sentence: "The incidence of POI increases with age..." is misleading since old women do not develop POI. I suggest to rephrase the sentence as follow: "the incidence of POI is highest in women aged from-to-".

2) The "conclusion" section is too long, contains repetitions of what is written in the text, and needs to be shortened.

Author Response

Response to Reviewer 1 Comments

Point 1: In the "abstract" the sentence: "The incidence of POI increases with age..." is misleading since old women do not develop POI. I suggest to rephrase the sentence as follow: "the incidence of POI is highest in women aged from-to-".

Response 1: Thank you. The abstract has been revised to specify that the incidence of PO increases with age in reproductive-aged women. The suggested rephrasing has been incorporated into the abstract, on page 1 line 12-13.

Point 2: The "conclusion" section is too long, contains repetitions of what is written in the text, and needs to be shortened.

Response 2: Thank you. We agree the conclusion was repetitive and not informative. The conclusion has been revised by significantly shortening the summary of fertility preservation options reviewed in the main text.

Reviewer 2 Report

Comment on the manuscript entitled:“Premature Ovarian Insufficiency: Procreative Management and Preventive Strategies”

The manuscript entitled “Premature Ovarian Insufficiency: Procreative Management and Preventive Strategies” is a review about the methods of fertility preservation for women affected with POI (Premature Ovarian Insufficiency). This review is well written and it is a good synthesis of these methods. I think this work very useful for physicians, biologists, and for teaching reproductive biology.

This article can be accepted for publication in biomedicines after some minor improvements.

General

The methods used for the management of procreation are well described. But not all of these methods are allowed in all countries. It would be useful to indicate this fact in a few words, or to specify from which country these data come (USA, I suppose).

Details

Page 6, lines 27 and 29: As2O3 instead As2O3

Page 6, end of line 33: add reference 53 which is given in the references list but certainly forgoitten in the text.

References:

Write the references according to the author instructions (verify the presence of useless italics, for instance), and check carefully.

Example:

Donnez, J. and M.M. Dolmans. Fertility Preservation in Women. N Engl J Med 2017, 377(17): p. 1657-1665.

Instead:

 Donnez, J. and M.M. Dolmans. Fertility Preservation in Women. N Engl J Med 2017, 377(17): p. 1657-1665

Author Response

Response to Reviewer 2 Comments

Point 1: The methods used for the management of procreation are well described. But not all of these methods are allowed in all countries. It would be useful to indicate this fact in a few words, or to specify from which country these data come (USA, I suppose).

Response 1: Thank you for your comment. We have addressed this point in the conclusion, on page 6 lines 44-49, and page 7 lines 3-9.

Point 2: Page 6, lines 27 and 29: As2O3 instead As2O3.

Response 2: Those changes have been made on page 6, lines 29 and 31.

Point 3: Page 6, end of line 33: add reference 53 which is given in the references list but certainly forgotten in the text.

Response 3: Thank you. That reference has been added on page 6, page 35.

Point 4: Write the references according to the author instructions (verify the presence of useless italics, for instance), and check carefully.

Response 4: Thank you. The references have been updated to the MDPI EndNote Style, and all references were reviewed for stylistic adherence to the author instructions. Unnecessary italics have been reviewed and corrected.